# Admissions to psychiatric inpatient services and use of coercive measures in 2020 in a Swiss psychiatric department: An interrupted time-series analysis

Alexandre Wullschleger[1]*, Leonel Gonçalves[2], Maya Royston[1], Othman Sentissi[1], Julia Ambrosetti[3], Stefan Kaiser[1], Stéphanie Baggio[4,5]

1 Department of Psychiatry, Geneva University Hospitals, Geneva, Switzerland, 2 Division of Prison Health, Geneva University Hospitals & University of Geneva, Geneva, Switzerland, 3 Department of Acute Medicine, Geneva University Hospitals, Geneva, Switzerland, 4 Institute of Primary Health Care (BIHAM), University of Bern, Bern, Switzerland, 5 Population Health Laboratory (#PopHealthLab), University of Fribourg, Fribourg, Switzerland

* alexandre.wullschleger@hcuge.ch

**Data Availability Statement:** All relevant data are within the paper and its Supporting information files.

## Abstract

### Background

The CoVID pandemic and the associated lockdown had a significant impact on mental health services. Inpatient services faced the challenge of offering acute psychiatric while implementing strict infection control measures. There is, however, a lack of studies investigating the use of coercive measures during the pandemic and their relation to hospitalizations and symptom severity.

### Aims

To investigate the effects of the CoVID outbreak on psychiatric admissions, use of seclusion and symptom severity.

### Method

Using routine data from 2019 and 2020 gathered in the Department of Psychiatry at the Geneva University Hospitals, we performed an interrupted time series analysis. This included the number of psychiatric hospitalizations, the proportion of people who experienced seclusion and the average severity of symptoms as measured by the Health of Nations Outcome Scale (HoNOS). Dependent variables were regressed on the time variable using regression model with bootstrapped standard errors.

### Results

Hospitalizations decreased over time ($b$ = -0.57, 95% CI: -0.67; -0.48, $p$ < .001). A structural break in the data (supremum Wald test: $p$ < .001) was observed in the 12th week of 2020. There was an inverse relationship between the number of admissions and the proportions of people subject to seclusion ($b$ = 0.21, 95% CI: -0.32; -0.09, $p$ < .001). There was a

**Funding:** AW and SK received funding from the Private Fondation of the HUG (https://www.fondationhug.org/en). The funders had no role in study design, data collection and analysis, decision to publish, or preparation of the manuscript.

**Competing interests:** The authors have declared that no competing interests exist.

statistically marginally significant inverse relationship between HoNOS scores at admission and the number of psychiatric hospitalizations ($b$ = -1.28, 95% CI: -2.59, 0.02, $p$ = .054).

## Conclusion

Our results show that the CoVID pandemic in 2020 was associated with a significant decrease in the number of hospital admissions. This decrease was correlated with a greater use of seclusion. The higher burden of symptoms and the difficult implementation of infection control measures might explain this higher use of coercion.

## Introduction

The CoVID outbreak had a significant impact on the mental health of the population. This was particularly true for people with psychiatric disorders [1]. Besides the psychological impact of the pandemic, they were particularly vulnerable from a somatic perspective, as studies have shown increased mortality risk in case of infection [2]. Mental health services, both inpatient and outpatient, therefore faced the challenge of adapting to this situation [3]. Scientific literature showed that the number of admissions to psychiatric emergency and inpatient services [4–7]. Similar findings were observed in non-psychiatric services [8].

The Swiss authorities imposed a national lockdown on March 16, including the closure of schools and most public places. The measures were then gradually lifted between April 27 and June 5, 2020. In public transport, the wearing of masks remained compulsory. In October 2020, with the start of the second pandemic wave in Switzerland, the authorities introduced a new set of measures. They generalized the wearing of masks in all public places and closed restaurants and most cultural and sports venues. Schools remained open during the second wave.

The lifting of the lockdown seems to have been associated with an increase in emergency room visits and hospital admissions [9]. These phenomena are undoubtedly a reflection of the huge impact of the lockdown and the pandemic on access to mental health care. Hospital services, including psychiatric services, had to adapt rapidly to public health and infection prevention requirements [6]. Therapeutic group activities were on hold, and visits and movements of patients in and outside the wards were restricted. Virtual meetings and get-togethers were quickly set up, but are unlikely to be a complete substitute for face-to-face contact, even if generally well accepted [10]. Strict control measures were introduced. These included systematic testing, isolation in the event of symptoms or proven infection, and interpersonal distancing [11]. Implementation of these procedures required great adaptability from staff [12–14]. Furthermore, the difficulties some mentally ill patients experienced in understanding and following the measures made them even harder to implement [15].

### Coercive measures

Coercion in mental health care is a complex phenomenon encompassing a wide range of practices, from informal pressures to measures restricting the liberty of inpatients [16, 17]. Some studies have reported an increase in the use of seclusion or mechanical restraint during the pandemic and, more generally, an increase in the number of violent incidents [18–21]. Conversely, other studies showed a decrease in mechanical restraint during this period [22, 23]. These works used pre-post statistical comparisons to assess the effect of the pandemic on these outcomes. The Swiss Association for the Quality of Hospital Care (ANQ) reported a national increase in the proportion of inpatients subjected to these measures in Switzerland in 2020

[24]. The detrimental effects of seclusion and restraint are well known, both in terms of their impact on patient satisfaction and their physical and psychological health (e.g. post-traumatic stress symptoms) [17]. Their potential increased use during the pandemic thus raises ethical and legal questions, particularly regarding the legal framework for coercive measures applied solely to prevent potential contamination in hospital settings [25, 26].

### Aims

The primary aims of the present study were twofold: (1) to test whether there were differences in (a) the number of psychiatric hospitalizations and (b) the proportion of people subjected to seclusion in 2019 and 2020 (before and during the pandemic); (2) to analyse the relationship between the number of psychiatric hospitalizations and the proportion of people subjected to seclusion during this period. As a secondary objective, we tested whether the number of psychiatric hospitalizations was associated with the severity of symptoms at admission. This study is the first to analyse the use of seclusion during the CoVID pandemic in direct relation to the number of hospitalizations and variations in symptom levels using a time series analysis method. This method has previously been used to analyse the impact of the pandemic on the use of mental health services[27, 28]. However, it has never been used in relation to coercive measures.

## Materials and methods

### Study design

The study was designed as a retrospective observational study.

### Data collection

Routine data were automatically retrieved from the clinic information system. Patients admitted to the Department of Psychiatry of the Geneva University Hospital between January 1, 2019, and December 31, 2020, were included.

The authors assert that all procedures contributing to this work comply with the ethical standards of the relevant national and institutional committees on human experimentation and with the Helsinki Declaration of 1975, as revised in 2008. The ethics committee of the Canton of Geneva approved the study protocol (no. 2021–00263, approved on February 25, 2021). The use of anonymized routine data didn't require the consent of patients.

### Variables

Variables included the number of psychiatric hospitalizations (voluntary and involuntary), the proportion of people who experienced seclusion (i.e., the number of seclusion measures divided by the number of psychiatric hospitalizations), and the average severity of symptoms as measured by the Health of Nations Outcome Scale (HoNOS). The HoNOS is a 12-item scale rating symptom severity and burden on a 0 to 4 scale. Items scores are added to build a total score ranging from 0 to 48. Higher scores indicate higher symptom severity [29, 30]. These variables were measured over time, corresponding to weekly values from week 1 of 2019 until week 52 of 2020, making a sample of 104 observations for analyses. Study variables by year are presented in the appendix (S1 Appendix).

Psychiatrists prescribe seclusion for initial 24 hours and can renew the prescription as judged necessary. Seclusion can only be applied to involuntarily admitted patients.

## Statistical analyses

Time series analyses were used, as the data consisted of successive measurements of the same source over time. To address the primary objective 1 (a) hospitalizations and (b) seclusion over time, as we were interested in measuring the trend in the variables, the dependent variables were regressed on the time variable using a linear regression model with bootstrapped standard errors. We tested whether the regression coefficients were stable over time with the test for unknown structural break (supremum Wald test [31]). The time series were graphically represented using the Hodrick-Prescott filter.

To address the primary objective 2 (association between psychiatric hospitalizations and the proportion of seclusion measures), because both variables appeared to be non-stationary based on the augmented Dickey-Fuller unit-root test [32] and the Kwiatkowski, Phillips, Schmidt, and Shin (KPSS) test for stationarity [33], they were transformed into first-order differences. Then, auto-regressive integrated moving average models (ARIMAX) with robust standard errors and including different autoregressive (AR) and moving average (MA) components were compared. The Bayesian Information Criterion (BIC) served to choose the model that best fitted the data. The analyses were also conducted on data from 2019 and 2020 separately to check the reliability of the estimates over time. Post-estimations were computed to test the statistical assumptions of stationarity (parametric autocorrelation function [34]), eigenvalue stability (inverse roots of the MA polynomial [35]), uncorrelated random variables (Portmanteau [36] and Bartlett [37] tests for white noise), parameter stability [31, 38], and co-integration (Engle-Granger co-integration test [39]). The predictions of the model were also presented in a graph. ARIMAX models with robust standard errors were also used to address the secondary objective (association between the number of psychiatric hospitalizations and the severity of symptoms at admission).

There were no missing values in the data. Statistical significance was set at $p < .05$. All analyses were conducted in Stata 17.

## Results

### Study population

Overall, 4768 hospital stays concerning 3175 patients were included in the analysis. In 2019, 2603 hospital admissions for 1977 patients were analysed. As to 2020, 2165 hospital stays concerning 1648 patients were included.

As to seclusion, it was used in 2019 in 358 hospital stays (13.8%) (323 patients), and in 370 stays (17.1%) (321 patients) in 2020.

As to symptoms severity, the mean HoNOS score at admission was 23.50 ($\pm$7.63) in 2019 and 23.87 ($\pm$6.73) in 2020.

### Psychiatric hospitalizations and proportion of seclusion measures over time

The number of psychiatric hospitalizations over time is presented in Fig 1. The number of psychiatric hospitalizations has a positive linear ($b = 0.56$, 95% CI: 0.12; 0.99, $p < .001$) and a negative quadratic trend ($b = -0.01$, 95% CI: -0.015; -0.007, $p < .001$). That is, the time series increased and then decreased over time. There was a structural break in the data (supremum Wald test: $p < .001$) in the 12[th] week of 2020.

The proportion of people subject to seclusion over time is presented in Fig 2. The proportion of people experiencing seclusion has a negative linear ($b = -0.52$, 95% CI: -0.77; -0.27, $p < .001$) and a positive quadratic trend ($b = 0.006$, 95% CI: 0.004; 0.008, $p < .001$). The time series

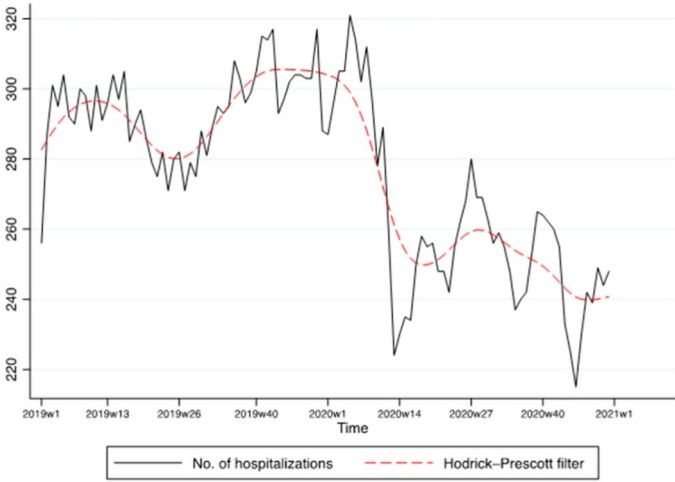

**Fig 1. Time series of the number of hospitalizations with Hodrick-Prescott filter.**

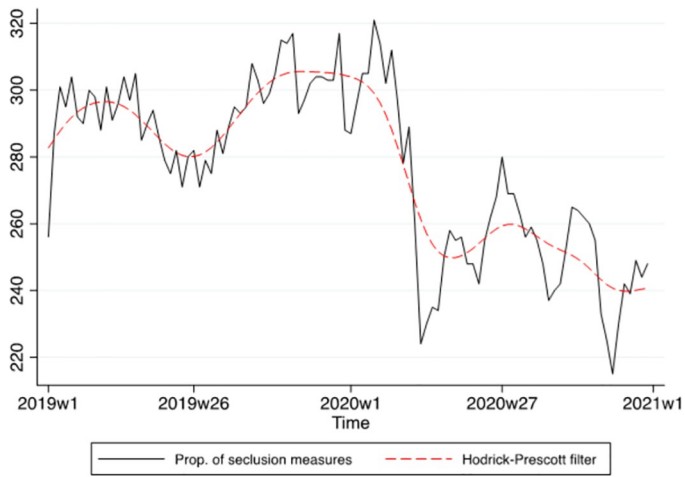

**Fig 2. Time series of the proportion of seclusion measures with Hodrick-Prescott filter.**

decreased then increased over time. There was a structural break (supremum Wald test: $p <$ .001) around the 14th week of 2020.

### Relation between psychiatric hospitalizations and the proportion of people subject to seclusion

The number of psychiatric hospitalizations and the proportion of people with coercive measures are presented in Fig 3. Since the beginning of 2020 (2020 line), there is an inverse relationship between these variables, with an increase in the proportion of seclusion measures when the number of hospitalizations decreased.

Based on the BIC for the selection of the model regressing the proportion of seclusion measures on the number of psychiatric hospitalizations, the ARIMAX model with 0 AR lags and a MA window of 1 (0, 1, 1 model) appeared to fit the data best. The regression analysis (see Table 1) revealed that the proportion of people facing seclusion measures and the number of

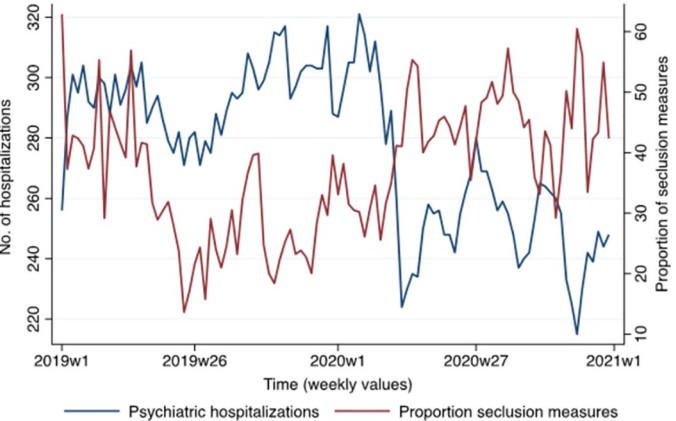

**Fig 3. Psychiatric hospitalizations and proportion of seclusion measures over time.**

psychiatric hospitalizations (analysed in first-order differences) were negatively related ($b$ = 0.21, 95% CI: -0.32; -0.09, $p$ < .001). That is, as the number of hospitalizations decreased, the proportion of people subject to coercion tended to increase. The moving average term was also significant ($b$ = -0.56, 95% CI: -0.74; -0.39, $p$ < .001). When running the analysis for the year 2019 only, there was no association between the variables ($b$ = -0.14, 95%CI: -0.41; 0.13, $p$ = .314). However, the association was significant when analysing the year 2020 only ($b$ = -0.24, 95%CI: -0.35; -0.14, $p$ < .001).

Post-estimations indicated that the residuals of the regression didn't deviate from white noise (Portmanteau test: $p$ = .363; Bartlett statistic: $p$ = .601), the parametric autocorrelations decayed exponentially toward zero suggesting stationarity, and the MA parameter satisfied the invertibility condition (eigenvalue = .56). Additionally, there was no evidence of a structural break in the data (supremum Wald test: $p$ = .174; cumulative sum test for parameter stability not significant). The Engle-Granger test confirmed that the study variables were cointegrated ($p$ < .001). The predictions of the model are presented in Fig 2 (see also S1 Appendix). The predicted and observed values were close, and most observed values were within the 95% confidence interval of the predictions confirming that the model fits the data well.

### Relation between psychiatric hospitalizations and severity of symptoms at admission

Regarding the association between the number of psychiatric hospitalizations and the severity of symptoms at admission, the ARIMA model measured on differentiated variables and with

**Table 1. Association between the number of psychiatric admissions and the proportion of seclusion measures over time.**

| Outcome: proportion seclusion measures | $b$ | SE | 95% CI | $z$ | $p$ |
|---|---|---|---|---|---|
| D1. Hospitalizations | -0.21 | 0.06 | -0.32, -0.09 | -3.52 | < .001 |
| cons. | -0.11 | 0.30 | -0.69, 0.48 | -0.36 | .720 |
| L1.ma | -0.56 | 0.09 | -0.74, -0.39 | -6.46 | < .001 |
| sigma | 6.93 | 0.50 | 5.96, 7.91 | 14.00 | < .001 |

*Note.* $b$ = unstandardized regression coefficient, SE = standard error, CI = confidence interval, $z$ = test of statistical significance, $p$ = significance level, D1 = difference, L1 = lagged value, ma = moving average term, cons. = constant term (intercept). $N$ = 103.

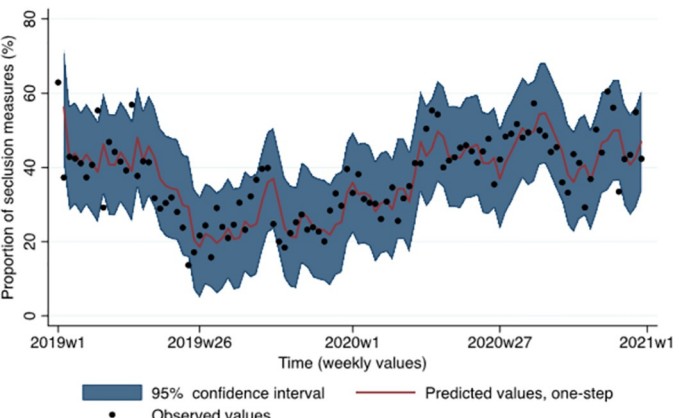

**Fig 4. Proportion of seclusion measures over time as a function of psychiatric admissions.**

no AR or MA components (0, 1, 0 model) fitted the data best. The results showed that symptom severity was inversely related with psychiatric hospitalizations, indicating that the severity of patients' symptoms was higher in periods when there was a lower number of admissions in the hospital. However, this association was only marginally significant at the 5% level ($b = -1.28$, 95% CI: -2.59, 0.02, $p = .054$). The proportion of seclusion measures with predictive values is presented in Fig 4.

## Discussion

The present study shows that the CoVID-19 pandemic was associated with a significant decrease in the number of hospital admissions during the first lockdown. This decrease was followed by an increase in admissions after the first CoVID wave, then by a further decrease during the second wave. The observed reduction in admissions correlated with an increase in symptom severity at admission and an increase in the number of seclusion measures used.

Fluctuations in the number of psychiatric admissions have been observed in previous studies [5, 9]. The decrease of admissions is certainly related to the strict admission policy enforced in the pandemic context. Due to infection control measures, the number of beds had to be limited to limit overcrowding of wards, and indications for psychiatric admission had to be strictly reviewed to limit the risk of spreading infection. The fact that all outpatient services remained open and continued to receive patients physically certainly contributed to this reduction. However, we can also hypothesize that people were reluctant to seek medical help during the first lockdown because of the contamination risk and the fear of being infected in the hospital. To some extent, this decrease may also be related to the fact that many people suffering from mood or anxiety disorders felt paradoxically relieved because of the imposed restrictions on social and work interactions.

After the initial decrease in psychiatric admissions during the lockdown period, a slight increase in admissions was observed, especially after the first lockdown measures were lifted. This can be seen as the first sign of the pandemic's adverse effects on the population's mental health, as shown by other studies [40]. The prolonged interruption of psychosocial services and the lack of contact with relatives had a severe impact on the health of people with severe mental illnesses, in turn partly explaining why the number of admissions increased again. Interestingly, however, the number of admissions hasn't returned to pre-pandemic levels, partly because infection prevention measures and services reorganization were maintained. However, this dynamic of an increase following the initial decrease, with structural measures

held constant, suggests that some specific factors related to patients' help-seeking behaviour played a role during the different phases of the pandemic.

Our second main finding suggests that the decrease in psychiatric hospitalizations was associated with an increase in the proportion of patients subject to seclusion, particularly in 2020. Since the COVID-19 pandemic (week 12 of 2020), the proportion of patients subjected to seclusion was above the mean (across 2019 and 2020) and the number of hospitalizations was below the mean. Recent scientific literature has shown mixed results on this outcome during the pandemic. Some studies seem to confirm this upward trend in the use of coercive measures, while others show a decrease during the pandemic [18, 23].

Several factors may explain the observed increase in the use of seclusion in our sample. First, the observed increase in symptom severity on admission during the pandemic suggests that only the most severely ill patients were admitted during the lockdown, which is congruent with the enforced strict admission policy. Symptom severity as measured by the HoNOS, especially the intensity of aggressive behaviour, is a significant risk factor for the use of coercive measures [41]. The observed increase in the proportion of secluded patients might thus reflect the higher severity of symptoms at admission.

A second factor that may explain our results is the fact that many patients, especially the most severely ill, had difficulties to comply with the infection control measures introduced on the wards [42]. As mentioned above, restrictions were imposed that were similar to those imposed in all services of the hospital. These restrictions had a significant impact on the wards' daily routine and on patients, whose freedom of movement was significantly restricted compared to the usual wards' policy. As a result, coercive measures were sometimes used to prevent the spread of infection on wards when patients refused mandatory CoVID testing on admission or presented with symptoms compatible with infection and were unable to comply with isolation measures. Besides, the fact that most psychiatric wards in our department apply an open-door policy, which was not suspended during the pandemic, may have contributed to this difficulty. With the very high epidemic pressure experienced in Geneva in 2020 [43], there was a high probability of contamination in the community, justifying the strict implementation of restrictive measures in the wards. This in turn may have led to greater use of seclusion to avoid contamination. A qualitative examination of the documented justification for seclusion would help to confirm this hypothesis and should be the subject of further research.

The present study is, to our knowledge, one of the first to present a relationship between the number of psychiatric admissions and the use of seclusion during the CoVID pandemic. Strengths of this work are the large sample size, covering the whole of 2020 and not just the first wave of the pandemic, and the sound methodological approach. Thus, the presented results provide an important insight into the changes in mental health services during the pandemic.

In terms of limitations, our analysis did not include other clinical and socio-demographic factors describing the population of patients admitted to inpatient services, such as gender, age, secondary diagnoses, or admission status. Such variables could not be included in the time series analysis. A more detailed analysis of the evolution of clinical and socio-demographic patients' characteristics during the pandemic at the individual level could provide further interesting information about psychiatric care during this period. The study did also not include data on staff, such as staff-patient ratios or the number of CoVID-related absences. Such information should be included in further research to assess factors contributing to the use of seclusion. Regarding seclusion measures themselves, data on their duration and justification were not included in the present study. It should also be noted that these results relate only to inpatient services and therefore do not provide information about changes in outpatient services, which certainly had an impact on inpatient services.

The results open new perspectives for further research. A follow-up analysis of subsequent pandemic waves and their impact on inpatient services would be interesting in terms of the adaptability of these services to a pandemic crisis, especially regarding the issue of coercive measures. Such an analysis would also provide important insights into the impact of the pandemic on the mental health of the population. On a clinical level, this study could help to develop institutional policies for dealing with similar pandemics and to adapt coercion reduction programmes to such contexts.

In conclusion, the present work shows that the CoVID pandemic was associated with a significant decrease in the number of psychiatric hospitalizations, especially during the first lockdown. This decrease was associated with greater symptom severity among admitted patients and increased use of seclusion. These results therefore provide an important insight into the impact of the pandemic on the organization of psychiatric services. They also highlight the need for further organizational measures to reduce seclusion [44], not only to counteract its negative effects, but also to limit the harmful effects of particularly stressful contexts such as the pandemic.

## Supporting information

**S1 Appendix. Study variables by year.**
(DOCX)

**S1 Dataset.**
(XLSX)

**S2 Dataset.**
(DO)

## Author Contributions

**Conceptualization:** Alexandre Wullschleger, Stefan Kaiser, Stéphanie Baggio.

**Data curation:** Alexandre Wullschleger, Leonel Gonçalves, Maya Royston, Stéphanie Baggio.

**Formal analysis:** Leonel Gonçalves, Stéphanie Baggio.

**Funding acquisition:** Alexandre Wullschleger, Stefan Kaiser.

**Investigation:** Julia Ambrosetti.

**Methodology:** Stéphanie Baggio.

**Project administration:** Alexandre Wullschleger.

**Supervision:** Stefan Kaiser.

**Visualization:** Stéphanie Baggio.

**Writing – original draft:** Alexandre Wullschleger.

**Writing – review & editing:** Alexandre Wullschleger, Leonel Gonçalves, Maya Royston, Othman Sentissi, Julia Ambrosetti, Stefan Kaiser, Stéphanie Baggio.

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
