## [Decision Letter · Decision Letter 0]

7 May 2023

PONE-D-23-02930

Admissions to psychiatric inpatient services and use of coercive measures in 2020 in a Swiss psychiatric Department: an interrupted time-series analysis

PLOS ONE

Dear Dr. Wullschleger,

Thank you for submitting your manuscript to PLOS ONE. After careful consideration, we feel that it has merit but does not fully meet PLOS ONE’s publication criteria as it currently stands. Therefore, we invite you to submit a revised version of the manuscript that addresses the points raised during the review process.

We look forward to receiving your revised manuscript.

Kind regards,

Jahida Gulshan

Academic Editor

PLOS ONE

Journal Requirements:

Reviewers' comments:

Reviewer's Responses to Questions

**Comments to the Author**

1. Is the manuscript technically sound, and do the data support the conclusions?

Reviewer #1: Yes

Reviewer #2: Partly

2. Has the statistical analysis been performed appropriately and rigorously? 

Reviewer #1: I Don't Know

Reviewer #2: No

3. Have the authors made all data underlying the findings in their manuscript fully available?

Reviewer #1: Yes

Reviewer #2: No

4. Is the manuscript presented in an intelligible fashion and written in standard English?

Reviewer #1: Yes

Reviewer #2: Yes

5. Review Comments to the Author

Reviewer #1: Review Report

This paper may be recommended for publication subject to the following minor modifications/corrections:

1. The novelty of the research must be clearly shown at the end of the introduction section.

2. The innovation should be clear in abstract. Improve it.

3. All assumptions and limitations for the current study should be stated point-wise.

4. Improve the paper grammar wise

5. What are the limitations of the defined problem?

6. Outlook and further perspective can be included in the conclusion part.

7. Literature review is also to be enriched with recent articles.

Reviewer #2: The current article investigated the effects of COVID outbreak on psychiatric admission. By using the data gathered over 2019-20 in the department of psychiatry at the Geneva University Hospitals, an interrupted time series analysis was performed, and discussion is made around the number of hospitalisations during COVID pandemic being correlated with a greater use of seclusion and higher use of coercion. This article is based on an important topic and can contribute in knowing the pattern of mental health associated with COVID outbreak and use of coercive measures, the study still needs improved presentation of results and overall strengthen the methodology part.

Introduction:

a) Though a number of published article was cited in Introduction, I believe some more relevant articles culd be reviewed, specifically the choice of analysis method could be supported by evidence from literature. If not changing or improving the method, at least some review of what other analysis methods have been used in similar datasets and to fulfill similar objectives could be reported. For instance, please see the articles below:

1. Aragonés-Calleja, M., & Sánchez-Martínez, V. (2022). Current State of Research on Coercion in Mental Health: Umbrella Review Protocol. Journal of Psychosocial Nursing and Mental Health Services, 60(10), 49-55.

2. Wiegand, H. F., Bröcker, A. L., Fehr, M., Lohmann, N., Maicher, B., Röthke, N., ... & Adorjan, K. (2022). Changes and Challenges in Inpatient Mental Health Care During the First Two High Incidence Phases of the COVID-19 Pandemic in Germany–Results From the COVID Ψ Psychiatry Survey. Frontiers in psychiatry, 13.

b)The rationale of the study needs to be more clearly stated. How will this study help to contribute, is there anything novel about the study, what will be the impact?

Methods:

While introducing he data and variables, some tabular and/or graphical representations of the variables, their interrelationships should be presented as part of the exploratory analysis. The choice of analysis method should be informed by the result of the exploratory analysis. I suggest adding these in this section.

Results:

While the result has been described in words using the coefficient values and P values, I still felt some tabular representation of all the necessary coefficients, their standard errors, P values will be more accessible to readers and will complement the descriptions made.

In the text, it is mentioned that ARIMA models showed inversed relationship between symptom severity and psychiatric hospitalizations, however, there is no table or figure to support the statement, apart from values mentioned in text line 207. I would recommend the author to review few papers to see how adequately model results are presented using tables and figures to strengthen the interpretation and revise the presentation of results in this article accordingly.

6. PLOS authors have the option to publish the peer review history of their article (what does this mean?). If published, this will include your full peer review and any attached files.

Reviewer #1: No

Reviewer #2: No

---

## [Author Response · Author response to Decision Letter 0]

2 Jun 2023

Dear Sir or Madam,

We thank you for the opportunity you gave us to submit a revised version of our manuscript. The reviewers’ comments helped us improve the quality of our work.

As to the general editor’s comments, we adapted the manuscript to the PLOS ONE style templates. The reference list was also enriched through the revision process. As to the availability of data, after checking with the local ethics committee, we are now able to provide the raw data and code as supplementary material.

Following the reviews for our manuscript, we are submitting a new version of the present work. The manuscript has been thoroughly revised and modified according to the very valuable comments made by the reviewers. All aspects and questions raised by the reviewers have been addressed, as stated below.

Answers to the reviewers’ comments

Reviewer #1: 

This paper may be recommended for publication subject to the following minor modifications/corrections:

1. The novelty of the research must be clearly shown at the end of the introduction section.

We thank the reviewer for his/her general appraisal of the paper. The introduction has been modified according to this comment.

2. The innovation should be clear in abstract. Improve it.

The abstract has been modified accordingly, stating more clearly the innovation of the present work.

3. All assumptions and limitations for the current study should be stated point-wise.

The issue of coercion in psychiatric care has been contextualized in the introduction. The limitations section has been completed and should now be clearer.

4. Improve the paper grammar wise

The manuscript underwent a thorough correction process.

5. What are the limitations of the defined problem?

The limitations section has been modified and completed, as mentioned above.

6. Outlook and further perspective can be included in the conclusion part.

The conclusion now includes a paragraph on further scientific and clinical perspectives.

7. Literature review is also to be enriched with recent articles.

According to the reviewer’s comment, the literature review now includes more recent papers.

Reviewer #2: 

1. The current article investigated the effects of COVID outbreak on psychiatric admission. By using the data gathered over 2019-20 in the department of psychiatry at the Geneva University Hospitals, an interrupted time series analysis was performed, and discussion is made around the number of hospitalisations during COVID pandemic being correlated with a greater use of seclusion and higher use of coercion. This article is based on an important topic and can contribute in knowing the pattern of mental health associated with COVID outbreak and use of coercive measures, the study still needs improved presentation of results and overall strengthen the methodology part.

We thank the reviewer for his/her positive general appraisal of the manuscript and his/her useful comments.

2. Introduction:

a) Though a number of published article was cited in Introduction, I believe some more relevant articles culd be reviewed, specifically the choice of analysis method could be supported by evidence from literature. If not changing or improving the method, at least some review of what other analysis methods have been used in similar datasets and to fulfill similar objectives could be reported. For instance, please see the articles below:

1. Aragonés-Calleja, M., & Sánchez-Martínez, V. (2022). Current State of Research on Coercion in Mental Health: Umbrella Review Protocol. Journal of Psychosocial Nursing and Mental Health Services, 60(10), 49-55.

2. Wiegand, H. F., Bröcker, A. L., Fehr, M., Lohmann, N., Maicher, B., Röthke, N., ... & Adorjan, K. (2022). Changes and Challenges in Inpatient Mental Health Care During the First Two High Incidence Phases of the COVID-19 Pandemic in Germany–Results From the COVID Ψ Psychiatry Survey. Frontiers in psychiatry, 13.

Answer: The introduction now includes more recent papers contextualizing the present research. This also includes methodological aspects of previous works and articles using a similar approach as the one used in this wok.

b) The rationale of the study needs to be more clearly stated. How will this study help to contribute, is there anything novel about the study, what will be the impact?

Answer: The introduction and conclusion sections now state more clearly the rationale of the study and its anticipated impacts.

3. Methods:

While introducing he data and variables, some tabular and/or graphical representations of the variables, their interrelationships should be presented as part of the exploratory analysis. The choice of analysis method should be informed by the result of the exploratory analysis. I suggest adding these in this section.

Answer: We added graphical representations of the variables to the figures. The graphical representation of hospitalizations is presented in Figure 1, of coercive measures in Figure 2, and of the relationship between hospitalizations and coercive measures in Figure 3.

We chose during study conception to use pre-planned analyses in the study that were in our opinion optimally designed to answer our research question. We wanted to avoid the risk of confirmatory analyses that might increase when choosing a data-driven analysis process.

4. Results:

While the result has been described in words using the coefficient values and P values, I still felt some tabular representation of all the necessary coefficients, their standard errors, P values will be more accessible to readers and will complement the descriptions made.

Answer: For the objective 2 (association between the number of psychiatric hospitalizations and the proportion of coercive measures), we added a table presenting the results of the regression output. Please see Table 2: Association between the number of psychiatric admissions and the proportion of coercive measures over time.

In the text, it is mentioned that ARIMA models showed inversed relationship between symptom severity and psychiatric hospitalizations, however, there is no table or figure to support the statement, apart from values mentioned in text line 207. I would recommend the author to review few papers to see how adequately model results are presented using tables and figures to strengthen the interpretation and revise the presentation of results in this article accordingly.

Answer: This was presented in Figure 3. However, the values were presented as standardized values, to make the scale of the different variables equivalent. In the new version of the manuscript, we changed Figure 3, now using the variables in their original scale. Please see Figure 3: Psychiatric hospitalizations and proportion of coercive measures over time. 

In addition, we added a Table were the values on the study variables (number of hospitalisations, proportion of coercive measures, and severity of symptoms) by year are presented. The table includes the observed values, plus the predicted proportion of coercive measures along with their 95% CI based on the results from the regression analysis. Please see Table 1: Study variables by year.

Finally, we added a figure illustrating the proportion of coercive measures over time, including observed and predicted values, based on the results from the regression model (using the number of psychiatric admissions as a predictor). Please see Figure 4: Proportion of coercive measures over time as a function of psychiatric admissions. This figure serves to confirm that the prediction model (objective 2) fits the data well. The values on the graph correspond to the values presented in Table 1: Study variables by year.

We sincerely hope that this new version will now suit the reviewers and the editors. We are looking forward to your feedback.

Best regards,

For the research team

Alexandre Wullschleger

---

## [Editor Report · Decision Letter 1]

6 Jul 2023

PONE-D-23-02930R1Admissions to psychiatric inpatient services and use of coercive measures in 2020 in a Swiss psychiatric Department: an interrupted time-series analysisPLOS ONE

Dear Dr. Wullschleger,

Thank you for submitting your manuscript to PLOS ONE. After careful consideration, we feel that it has merit but does not fully meet PLOS ONE’s publication criteria as it currently stands. Therefore, we invite you to submit a revised version of the manuscript that addresses the points raised during the review process.

We look forward to receiving your revised manuscript.

Kind regards,

Jahida Gulshan

Academic Editor

PLOS ONE

Journal Requirements:

**Additional Editor Comments:**

**1. Move table 1 to appendix.**

**2. A thorough grammar check is necessary.**

---

## [Author Response · Author response to Decision Letter 1]

12 Jul 2023

Dear Prof. Gulsham,

We thank you for the opportunity you gave us to submit again a revised version of our manuscript. 

We addressed the two comments you made. Table 1 has been moved to the Appendix, and the manuscript has been thoroughly checked regarding language and grammar.

We sincerely hope that this new version will now fully comply with the publication standards of PLOS One. We are looking forward to your feedback.

Best regards,

For the research team

Alexandre Wullschleger

---

## [Editor Report · Decision Letter 2]

17 Jul 2023

Admissions to psychiatric inpatient services and use of coercive measures in 2020 in a Swiss psychiatric Department: an interrupted time-series analysis

PONE-D-23-02930R2

Dear Dr. Wullschleger,

We’re pleased to inform you that your manuscript has been judged scientifically suitable for publication and will be formally accepted for publication once it meets all outstanding technical requirements.

Kind regards,

Jahida Gulshan

Academic Editor

PLOS ONE

Additional Editor Comments (optional):

Thank you very much for your revised version.
---

## [Editor Report · Acceptance letter]

20 Jul 2023

PONE-D-23-02930R2 

Admissions to psychiatric inpatient services and use of coercive measures in 2020 in a Swiss psychiatric Department: an interrupted time-series analysis 

Dear Dr. Wullschleger:

I'm pleased to inform you that your manuscript has been deemed suitable for publication in PLOS ONE. Congratulations! Your manuscript is now with our production department. 

Kind regards, 

on behalf of

Dr. Jahida Gulshan 

Academic Editor

PLOS ONE